# Skeletal Muscle Measurements in Pediatric Hematology and Oncology: Essential Components to a Comprehensive Assessment

**DOI:** 10.3390/children10010114

**Published:** 2023-01-05

**Authors:** Kelly Rock, Odessa Addison, Vicki L. Gray, Robert M. Henshaw, Christopher Ward, Victoria Marchese

**Affiliations:** 1Department of Physical Therapy and Rehabilitation Science, University of Maryland School of Medicine, Baltimore, MD 21201, USA; 2Department of Orthopedic Oncology, MedStar Washington Hospital Center, Washington, DC 20010, USA; 3Department of Orthopedic Oncology, Children’s National Health System, Washington, DC 20010, USA; 4Department of Clinical Orthopedic Surgery (Orthopedic Oncology), Georgetown University School of Medicine, Washington, DC 20057, USA; 5Departments of Orthopedics and Biochemistry & Molecular Biology, University of Maryland School of Medicine, Baltimore, MD 21201, USA

**Keywords:** muscle architecture, muscle performance, biomechanics, children, cancer

## Abstract

Children with hematologic and oncologic health conditions are at risk of impaired skeletal muscle strength, size, and neuromuscular activation that may limit gross motor performance. A comprehensive assessment of neuromuscular function of these children is essential to identify the trajectory of changes in skeletal muscle and to prescribe therapeutic exercise and monitor its impact. Therefore, this review aims to (a) define fundamental properties of skeletal muscle; (b) highlight methods to quantify muscle strength, size, and neuromuscular activation; (c) describe mechanisms that contribute to muscle strength and gross motor performance in children; (d) recommend clinical assessment measures; and (e) illustrate comprehensive muscle assessment in children using examples of sickle cell disease and musculoskeletal sarcoma.

## 1. Introduction

The screening and measurement of neuromuscular function is an essential component of the physical examination to identify impairment, provide appropriate referrals, and track changes due to medical and exercise interventions. Neuromuscular strength is the measure of an individual’s ability to exert maximal muscle force and produce joint torque statically or dynamically [1]. Strength assessment is routine in physicians’, nurses’, occupational therapists’, and physical therapists’ practice. The Guide to Physical Therapy Practice and the Movement Systems Diagnosis Framework identify muscle performance and force production deficit as key elements of the physical therapy examination [2,3,4] Tests and measures of muscle performance include muscle force, joint torque, power, and endurance [4].

Persistent deficits in neuromuscular force production may arise from primary deficits at the muscle, neuromuscular junction, peripheral nerve, or central nervous system. The deficits may affect a focal joint or multiple joints [2,3]. The transient deficit that arises from activity, and recovers with rest, is termed muscle fatigue [2,3]. Ideally, clinical assessments should include outcome measures that aid in identifying the underlying mechanisms that contribute to neuromuscular deficit. Here, we focus on children with hematologic and oncologic health conditions in which the disease condition and the medical treatment have negative impacts on neuromuscular function, particularly at the level of the skeletal muscle [5,6,7,8,9,10,11,12]. Understanding the underlying mechanisms that contribute to muscle strength and those that affect gross motor performance, will allow for the appropriate selection of objective assessment measurements and the development of targeted exercise interventions for children with hematologic and oncologic health conditions.

This review aims to (a) define the fundamentals properties of skeletal muscle; (b) highlight research and clinical methodology to quantify skeletal muscle strength, size, and neuromuscular activation; (c) describe contributors to skeletal muscle strength in children and adolescents; (d) recommend clinical assessment measurements of skeletal muscle; and (e) illustrate examples for comprehensive muscle assessment in children with hematological and oncological health conditions using sickle cell disease and musculoskeletal sarcoma diagnoses.

## 2. Skeletal Muscle Fundamentals

### 2.1. Anatomy and Physiology

Skeletal muscle is primarily responsible for voluntary and graded bodily movement [13]. The anatomic structure of skeletal muscle comprises muscle fascicles, bundles of long multi-nucleated muscle fibers encased in fibrous connective tissue. Each muscle fiber contains highly organized bundles of myofibrils whose periodic arrangement of thick filament myosin and thin filament actin contractile proteins, termed sarcomeres, defines the ‘striated’ appearance under the light microscope. Interdigitated with the sarcomeres is the specialized sarco-endoplasmic reticulum (SR) that stores calcium (Ca^2+^). needed for activation of the thick filament to elicit myosin contraction.

Through a process called excitation–contraction coupling, motor nerve action potentials arriving at the pre-synaptic motor end-plate are transduced into acetylcholine signals released into the neuromuscular junction (NMJ) synapse that generate action potentials in the muscle fiber. It is the rapid propagation of action potentials through invaginations of the muscle cell membrane (t-tubules) that activate membrane L-type Ca^2+^ channels which elicit Ca^2+^ release from ryanodine receptor Ca^2+^ channels in the SR to drive muscle fiber contraction. The relaxation of the muscle fiber is marked by the cessation of action potentials and the sequestration of Ca^2+^ back into the SR via the sarco-endoplasmic ATPase (SERCA).

Within the muscle, groups of muscle fibers are innervated by a motor neuron, together called a motor unit. Motor units vary in the number of fibers, which is inversely proportionate to the level of precision. Smaller motor units that consist of fewer muscle fibers perform precision actions such as finger and eye movements. On the other hand, larger motor units of many muscle fibers perform more powerful movements, for example the quadricep muscles of the legs. A group of motor units of varying sizes are innervated to provide coordinated contractions within a single muscle to produce the appropriate movement patterns required for gross motor and fine motor tasks.

Specific types of muscle contraction depend on changes in tension or in length of the muscle fibers [13]. The two main types of skeletal muscle contractions are (1) isometric—increased muscle tension but no change in muscle length; and (2) isotonic—no change in muscle tension but a change in muscle length. Isotonic contractions are further differentiated by concentric contractions in which the muscle length shortens and during eccentric contractions in which the muscle lengthens. Muscle contractions can also be described as isokinetic if the muscle force causes the joint to move at a constant speed. Gross motor tasks such as walking, running, and jumping require a complex interplay between graded muscle activation and orchestrated muscle contraction mechanics, such as isometric contractions that stabilize the torso and a combination of concentric and eccentric contractions that move and then brake the lower extremity joints, respectively.

### 2.2. Muscle Macrostructure

The architecture and anatomic size of skeletal muscle are key determinates of its function (i.e., force, shortening velocity and the extent of shortening [excursion]). Thus, the quantification of muscle architecture and anatomic size provide important insights into the gain or loss of function, and measurements are used to quantify muscle macrostructure. Muscle architecture is defined as the length of the muscle fibers and their physical arrangement relative to the tendon [14,15]. Given the resolution needed to visualize muscle fibers, fascicle length is often the surrogate measure for muscle fiber length. Evidence that the number of sarcomeres in series (i.e., muscle fiber length) is positively related to the velocity and excursion; therefore, a change in muscle fiber length can impact shortening velocity, excursion, or muscle power [14,15]. Pennation angle is the measured geometric angle of muscle fibers in relation to the long axis of the muscle/tendon [14,15]. Given that increased pennated muscle have an increased number, yet shorter, muscle fibers per unit volume, these muscles tend to be of greater mass, and produce more force, but have decreased shortening velocity and excursion than non-pennated muscles [14,15].

Muscle force production is correlated to the number of sarcomeres in parallel which is related to the number of muscle fibers in cross-section [14,15]. Therefore, the cross-sectional area of the muscle measured at its widest point (two dimensional) is often used as a measure of anatomic size. In an attempt to account for muscle fiber pennation, the physiological cross-sectional area (PSCA) is calculated as the cross-sectional area perpendicular to the muscle fibers. While this has been common practice, recent evidence suggests that muscle force measured in vivo is not significantly impacted by pennation angle [16].

While the gold standard measure of human skeletal muscle architecture is invasive whole muscle dissection or muscle biopsy, non-invasive measures such as magnetic resonance imaging (MRI), computed tomography (CT), and ultrasonography (US) have been validated to quantify pennation angle, muscle fiber and fascicle length, cross-sectional area, and muscle thickness (Figure 1). MRI, CT, US imaging and bioelectric impedance can also supply important information about muscle volumes. These imaging methods allow for muscle macrostructure to be assessed independently from or in response to muscle contraction and provide insight into muscle performance capacity.

### 2.3. Muscle Performance

Assessments of muscle performance are used to quantify muscle function and to track its change with disease, treatment, or training. Here, we focus on in vivo assessments in humans where muscle force is an indirect measure of a major muscle contracting across a joint (e.g., quadriceps via knee extension) to generate torque on a lever (i.e., tibia, with force measured at the distal tibia) or a more complex action of multiple muscles (e.g., handgrip). Isokinetic dynamometers can measure the amount of muscle force and joint torque during conditions when the joint does not move (isometric); moves at a constant speed (isokinetic); or moves with a constant tension (isotonic). Handheld dynamometers measure isometric muscle force and joint torque [17]. Maximal muscle force can also be measured using one-repetition max (1RM) testing, the greatest amount of weight that can be moved through an isolated single joint or a functional task such as handgrip or a leg press for one repetition but not for a second repetition [18]. In clinical practice, clinicians often use manual muscle testing to rate the level of maximal muscle force compared to an external force applied by the assessor [19]. Muscle power, the product of force and contraction velocity, provides information about the ability for explosive or ballistic movements and can be measured through isokinetic and isotonic contractions and timed, rapid functional tasks such as leg presses, sit-to-stands, jumping, or stair climbing [20,21,22]. Muscle endurance can be measured by examining muscle fatigue as it provides a measure of the muscle’s ability to sustain a contraction and force over a longer duration, often greater than 30 s. Muscle endurance or fatigue can be quantified after a prolonged or repetitive muscle contraction, typically isometric, with force or torque measurements to indicate the failure of muscle force or torque [22,23]. A summary of assessment techniques to measure muscle performance is outlined in Figure 2.

### 2.4. Neuromuscular Activation

The contraction of skeletal muscles is under volitional control initiated in the brain, often referred to as central control. Specifically, biochemical processes of voltage changes across cellular membranes (action potential propagation) and chemical synapses, carry signals from the central nervous system through motor neurons to the motor units to regulate skeletal muscle fiber contraction. Each motor unit acts as a group with a unique pattern of activation. Activation of more or fewer motor units causes graded muscle contraction and relaxation, respectively. This neuromuscular activation can be measured using electromyography (EMG), a technique that records motor unit electrical signals (action potentials). EMG, measured in a unit of volts, can quantify the number and intensity of muscle fibers contracting in relationship to a recording electrode, which can be placed on the surface of the skin over (surface EMG) or directly into (needle/fine wire EMG) the muscle of interest. Motor unit recruitment and firing frequency can be decomposed from multiple electrodes placed on the skin over the muscle (high-density surface EMG) or needle/fine wire EMG. Surface EMG, a non-invasive measurement of muscle fibers directly under the electrode, is commonly used in rehabilitation science as a measure of neuromuscular activation [24]. The amplitude of the EMG signal provides insight into the intensity and velocity of the muscle contraction and regulation of force, and the frequency domain of the EMG signal is a power tool for assessing muscle fatigue where there is a downward shift in frequencies with fatigue. Other neuromuscular activation measures including the ability to quickly (rate of activation) and selectively (co-contraction) activate skeletal muscles are important in understanding the complex muscle activation patterns required for gross motor performance.

## 3. Muscle Strength in Children and Adolescents

Skeletal muscle strength, a measure of an individual’s ability to exert maximum muscle force and produce joint torque statically or dynamically, is an important indicator of muscle performance [15]. Muscle strength of the lower extremity muscles is associated with the ability to perform gross motor skills [1,25,26]. Therefore, lower extremity muscle strength is an important element of the rehabilitation assessment to screen for muscle impairment such as muscle strength deficits or to measure changes in response to exercise interventions. Muscle strength impairments, also referred to as muscle weakness or force production deficits, are important to identify in children and adolescents given the relationship between muscle strength and motor performance [27,28,29,30,31,32,33,34,35,36].

### 3.1. Contributors to Muscle Strength

Muscle strength measurements provide important information about the muscle’s ability to produce force or torque; however, these measures provide limited information about the underlying mechanisms that contribute to muscle strength including muscle size, neuromuscular activation, and other biomechanical and developmental considerations. Therefore, to comprehensively measure muscle strength, these additional aspects of skeletal muscle properties and performance need to assess the underlying mechanism of strength production and muscle performance.

#### 3.1.1. Muscle Size

Studies investigating muscle size in healthy children describe moderate to strong correlations between skeletal muscle strength and power with the cross-sectional area, volume, and thickness measurements [37,38,39,40,41]. Specifically, muscle strength is positively correlated with cross sectional area (plantarflexion: *r* = 0.52 to 0.09; dorsiflexion: *r* = 0.41 to 0.76) [39], volume (knee extension: *r* = 0.47 to 0.58; plantar flexion: *r* = 0.53 to 0.70) [37], and thickness (knee extension: *R^2^* = 0.59 to 0.77) [41]. MRI, CT and US imaging measuring skeletal muscle volume, cross-sectional area, and muscle thickness have identified decreased muscle size in children with chronic health conditions such as cerebral palsy and hemophilia [27,28,29,30,31,32,33,34,35,36,42,43]. In children with cerebral palsy, the size of the lower extremity muscles is positively correlated with muscle strength [41,42,43] *R^2^* values up to 0.56 [41,43] for knee extension. Increased lower extremity size is associated with improved gross motor performance measures such as the Gross Motor Function Measure (GMFM) [30,31,35], the Pediatric Evaluation of Disability Inventory (PEDI) [35], and gait characteristics [32]. Thus, muscle size measured by noninvasive imaging can provide important information about muscle strength and gross motor performance in children with and without health conditions.

#### 3.1.2. Neuromuscular Activation

EMG is used to measure neuromuscular activation, such as the rate of muscle activation (RoA) and muscle co-contraction indices. These measures can assess the capacity to rapidly activate and selectively activate the muscles around a joint during isolated and functional movements [44,45,46].

##### Rate of Muscle Activation

The rate of muscle activation (RoA) is a neuromuscular measure dependent on speed and is important for maximum joint torque and the proficiency of motor skills [26,47]. RoA can be measured using surface EMG. Although the maximum torque production is limited by muscle size [48], maximal joint torque, even after normalizing for muscle size, increases with age in children [26,49,50,51]. Therefore, in addition to muscle size, other factors may influence the capacity to generate joint torque, and neuromuscular mechanisms may contribute to these observations in children.

In adults, those with a greater rate of lower extremity muscle activation generate more torque [52,53]. There is evidence that greater early quadriceps activation (i.e., within the first 250 ms) in young adults is related to the performance of gross motor tasks commonly used in sports requiring sprinting faster and jumping higher [54]. This time period occurs before peak torque production and may indicate that the ability to activate the muscles earlier and faster is necessary for the proficiency of gross motor skills. Neuromuscular activation differs between children and adults, with children demonstrating a lower rate of gastrocnemius muscle activation [55]. Differences in gastrocnemius activation were also demonstrated between younger and older children (age ranges: 5–6 years, 7–8 years, 9–10 years), with the older age group demonstrating a greater rate of muscle activation [55]. This suggests that the rate of muscle activation in children is an important factor that influences the early stages of torque development [55]. Lower rate of activation in younger children compared to older children and adults may also be indicative of neural maturation, which occurs from childhood through adolescence specifically through maturation of the central nervous system [55]. Children also demonstrate decreased capacity to activate type II fast muscle fibers, and may present with slower conduction in peripheral nerve, higher latency at the neuromuscular junction, and lower motor unit firing frequencies [55]. Therefore, the rate of muscle activation may provide important information related to muscle strength in children, especially during tasks that require the ability to rapidly activate the lower extremity muscles such as balance and agility tasks, and warrants further investigation.

##### Co-Contraction

Gross motor performance requires the ability to coordinate muscle contraction among multiple joints. Surface EMG can measure co-contraction, the simultaneous activity of agonist and antagonist muscles crossing the same joint but on opposite sides of the joint. Co-contraction indices provide information about the ability to coordinate muscle contraction with low co-contraction indices during isometric joint conditions indicative of selective control [56]. High co-contraction indices may signify difficulty with selective control, underdeveloped reciprocal inhibition, learning a new skill, or pathological or inefficient movement patterns [56,57]. In adults with Down syndrome [58], adults with neurological conditions such as stroke [59], and adults with orthopedic conditions such as chronic neck pain [60], co-contraction indices have been reported to be elevated. Additionally, compared to adults, children demonstrate higher levels of thigh co-contraction during gross motor tasks such as walking [61], and younger children presented with higher co-contraction indices compared to older children [62]. Higher levels of co-contraction during gross motor tasks decrease the efficiency of movement and increase the metabolic expenditure [62]. Damiano et al. (2000) [63] identified high thigh co-contraction in children with cerebral palsy and significant relationships between co-contraction indices during isometric strength testing and walking [63]. These studies suggest that the co-contraction index may be an important underlying mechanism that result in deficits of muscle strength and gross motor performance in children with and without health conditions.

#### 3.1.3. Other Biomechanical and Developmental Considerations

Muscle strength is also influenced by muscle fiber types, tendon elasticity, muscle quality, and central activation [49,64,65]. Compared to adults, children may present with fewer type II (fast-twitch) muscle fibers and more type I (slow-twitch) muscle fibers [64]. Since type II fiber motor units have faster contraction speeds, a lower ratio of fast-twitch muscle fibers may adversely contribute to the ability to produce muscle force or joint torque [47]. Inversely, the presence of increased ratios of type I fiber motor units increases endurance capacity in children compared to adults [23,66]. Overall, children have smaller muscle fibers and the size ratio between type I and type II muscle fibers are similar contrary to the expected adult size discrepancy pattern in which type II muscle fibers are larger than type I muscle fibers [67]. Another age-related change throughout childhood into adulthood is that tendon stiffness increases [47,68,69]. A summary of the expected child-adult differences in skeletal muscle are demonstrated in Figure 3. Given that the mechanical properties affect the transfer of force to joints, decreased tendon stiffness can reduce muscle force, joint torque, and rate of muscle activation measures [70]. Additionally, fat or adipose tissue infiltration into skeletal muscle, known as myosteatosis, is associated with muscle disease processes [71,72,73,74]. Using MRI, CT, and US imaging, skeletal muscle adipose tissue quantity and quality can be assessed. Adipose tissue infiltration has been identified in children with gross motor performance limitations with diagnoses such as cerebral palsy and muscular dystrophies [73,74]. Lastly, central activation measures the differences between force elicited during a maximal voluntary muscle contraction compared to the force elicited during a maximal voluntary muscle contraction with simultaneous external electrical stimulation of the muscle or motor nerve [47,75]. Evidence supports that children have lower volitional activation compared to adults, which may contribute to lower muscle force and power production compared to adults [47,76]. Volitional activation may also influence positive changes in force production in response to exercise training without changes in muscle size [76].

## 4. Clinical Assessment of Muscle

Current clinical objective assessment of muscle strength often focuses on maximum muscle force or joint torque measurements including manual muscle testing and dynamometry. Handheld dynamometry has moderate-to-good reliability and validity and is a more portable and affordable measurement to quantify muscle strength compared to “gold standard” isokinetic dynamometry (i.e., Biodex and Cybex systems) in children as young as 4 years of age [17,77,78,79,80]. Handheld dynamometry also has favorable psychometric properties compared to manual muscle testing [19]. Thus, the handheld dynamometer is likely the most appropriate and least expensive clinical tool for objective measurement of muscle strength in most clinical settings. Handheld dynamometry has established reliability and validity in children as young as 4 years of age [17,80]. The use of handheld dynamometry also allows for the assessment of muscle strength in a variety of testing positions and across body types, abilities and medical conditions. Since the level of voluntary activation is unknown, children should be provided with practice trials before formal measurement to familiarize the child with the task [20].

Noninvasive imaging assessments, such as US, can provide important measurements of muscle size. US can provide accessible, affordable, non-invasive reliable, and valid real-time muscle size measurements, including muscle thickness and anatomical cross-sectional area [14,17,29,81,82]. In healthy adults, quadriceps US muscle thickness has demonstrated concurrent validity with MRI measures of muscle thickness, volume, and cross-sectional area [83,84,85]. Additionally, changes in muscle thickness have been described as a consequence of post-operative changes, muscle wasting, and aging, demonstrating the clinical utility of this parameter in the examination of normal and impaired skeletal muscle [86,87,88]. Therefore, US measurement of muscle thickness may serve as a beneficial non-invasive clinical assessment tool in children and can be performed when medical restrictions prohibit strength testing, or when the child cannot follow instructions to complete a strength testing protocol.

Given that the ability to perform gross motor tasks requires muscle strength, as part of clinical pediatric rehabilitation, muscle strength should be not only assessed during single-joint movements but also during functional tasks, which require multi-joint movements. Many standardized gross motor functional tests exist for children, but tests that specifically address muscle performance include the Bruininks–Oseretsky Test of Motor Proficiency, second edition (BOT-2; ages 4 to 21 years) [89], the EUROFIT Test (ages 6 to 18 years) [90], the Functional Strength Measurement (ages 4 to 10 years) [22], and the Motor Performance Test (ages 4 to 11 years) [91]. These tests include gross motor tasks that require skeletal muscle force, power, endurance, rate of muscle activation, and coordination.

## 5. Muscle Considerations in Pediatric Hematology and Oncology

The comprehensive assessment of skeletal muscle is essential in children with hematologic and oncologic health conditions, especially given that these diseases and their medical treatments have negative effects on skeletal muscle. Survivors of childhood hematological and oncological health conditions, including sickle cell disease and musculoskeletal sarcoma, are at risk of impaired muscle strength. Therefore, exploration of muscle strength, muscle size, and neuromuscular activation and the relationships to gross motor performance is warranted.

### 5.1. Sickle Cell Disease

Sickle cell disease (SCD) is a genetically inherited condition that occurs in an estimated 2000 newborns in the United States per year, predominately of Sub-Saharan African but also Central and South American, Middle Eastern, Asian, and Mediterranean descent [92,93]. The progression of SCD occurs over the first six to 12 months of life in infants with SCD when fetal hemoglobin transitions to abnormal adult hemoglobin (S or C), causing polymerization of abnormal hemoglobin and irregularly shaped red blood cells. In children with SCD, the sickled form of hemoglobin (S) results in improper transportation of oxygen and blood flow, known as sickle cell anemia. The sickle-shaped red blood cells lack the flexibility needed to circulate in blood vessels, are fragile, have a shortened life span, and have increased adhesiveness to vascular endothelium. These adverse changes in the red blood cell can lead to vaso-occlusion in small blood vessels and local ischemia in body tissues, resulting in painful episodes, known as vaso-occlusive crises. Recurrent vaso-occlusive crises and chronic anemia can lead to long-lasting damage to body organs and tissues and create an inflammatory cascade, which can cause further tissue damage to the bones (avascular necrosis and osteomyelitis), muscle (myonecrosis), brain (cerebral infarction) and lungs (acute chest syndrome, pulmonary hypertension, and chronic lung disease) [5,93,94].

Current medical management for SCD that targets disease modification and symptom management includes pharmacological agents such as hydroxyurea and L-glutamine, and procedures such as blood transfusion and bone marrow transplant. Hydroxyurea, also known as hydroxycarbamide, increases total and fetal hemoglobin levels, lowers leukocyte levels, and decreases the expression of adhesive molecules on red blood cells, neutrophils, and vascular endothelium [95]. Hydroxyurea improves blood flow and reduces the number, frequency, and severity of vaso-occlusive crises [95]. L-glutamine is an FDA-approved supplement used to negate the increased demand for glutamine, a necessary amino acid, in states of stress caused by SCD [95]. Blood transfusions are commonly used to resolve acute and/or chronic anemia [95]. The only known cure for SCD is allogenic bone marrow transplantation, in which the recipient’s bone marrow is medically suppressed and replaced with a donor’s bone marrow [95]. The bone marrow transplantation process carries risks for graft rejection and infection and therefore is indicated only under specific circumstances [95].

Muscle changes due to vaso-occlusive crises have been shown to lead to muscle atrophy and contractures [5]. Additionally, mice with SCD demonstrate alterations in muscle properties causing impaired muscle performance such as lower electrically induced tetanic contractions, calcium-handling deficiencies, and impaired force relaxation and post-activation potentiation, which influence the effect of previous muscle contractions on subsequent contractions [5,6,7]. Thus, children and adolescents with SCD are at risk for impairments in neuromuscular activation. Impairments in lower extremity muscle strength, muscle size, and neuromuscular activation may affect gross motor performance in survivors of childhood SCD. However, to date, no studies have explored neuromuscular activation nor the relationships between muscle properties and gross motor performance in children with SCD [96].

Previous studies have identified impaired skeletal muscle strength in survivors of childhood SCD [96,97,98,99,100,101]. In studies comparing children with SCD with a control population, children with SCD presented with significantly decreased lower-extremity strength, maximal handgrip strength, ankle plantarflexion strength, and back strength [97,98,99,100,101]. Moheeb et al. (2007) [100] explored leg strength in children aged 9 to 12 years with SCD (*n* = 50), with sickle cell trait (*n* = 50), and healthy controls (*n* = 50). The authors report significantly lower leg muscle strength (*p* < 0.001) in children with SCD (23.3 ± 0.7 kg) compared to controls (28.1 ± 1.4 kg) [100]. Wali and Moheeb (2011) [101] explored leg strength in 93 children aged 10–14 years and reported decreased leg strength in children with SCD compared to healthy controls (*p* < 0.05). In the previous two studies [100,101], the authors mentioned the use of dynamometry, but methods of muscle testing, including testing position or muscle groups, were not reported [100,101]. Lastly, Doughtery et al. (2020) [99] measured knee extension strength of the left lower extremity using an isokinetic dynamometer at 60°/s. Children with SCD aged 5–20 years (*n* = 21) presented with lower peak knee extension torque (49.6 ± 27.6 Nm) compared to controls (*n* = 23; 51.7 ± 34.0 Nm), however, this comparison did not reach statistical significance [99].

Gross motor performance has been reported to be limited in children with SCD performing specific tasks required for participation in school, play, and sports activities [96,99,100,102,103,104,105,106,107,108]. Children with SCD demonstrate walking below norm-referenced distances on the 6 min walk test [102,103,104,105,106]. Children with SCD performed poorer on the 20-yard swim, 40-yard swim, 100-yard “potato race”, and jump height compared to age- and sex-matched controls [100,107]. The mean Bruininks–Oseretsky Test of Motor Proficiency (BOT) Short Form, a measure of motor performance, in children with SCD was reported as 61.5 to 67.4 and was similar to controls (62.3) [99,108]. However, as the BOT Short Form only has one running speed and agility measure (one-legged stationary hop) and two strength measures (push-ups and sit-ups), it is not comprehensive for assessment of the gross motor skills required for school, play, and sports activities as compared to the complete test of subscales. In adults with SS-genotype SCD, maximal knee extension torque was moderately correlated with gross motor performance on the 6MWT (*r* = 0.62), but no relationships between maximal EMG signal amplitude and gross motor performance [109]. However, the relationships among skeletal muscle strength, muscle size, neuromuscular activation, and gross motor performance have not been explored in child and adolescent survivors of childhood SCD. Thus, further studies objectively measuring skeletal muscle strength, muscle size, and neuromuscular activation and relationships between these variables and gross motor performance are needed in children with SCD. The identification of underlying skeletal muscle mechanisms such as myosteatosis are warranted.

### 5.2. Musculoskeletal Sarcoma

Musculoskeletal sarcomas (MSS), such as osteosarcoma and Ewing sarcoma, generally arise in the second decade of life from transformed mesenchymal connective tissue cells. MSS can affect any bone, but most commonly occur in long bones of the lower and upper extremities (i.e., femur, tibia, and humerus) or the pelvis [110]. Although MSS accounts for 3.4% of childhood cancer and affects approximately 750 children and adolescents per year in the United States, these children and adolescents comprise a patient population that has a high likelihood of recovery and survival [111,112]. Improvements in chemotherapeutic, radiation, and surgical management in child and adolescent survivors of MSS have increased long-term survivorship over the past few decades, with a 5-year overall survival rate reaching 70% [111,112].

Treatment for MSS has evolved with the primary goal of not only improving survival, but also preserving the individual’s long-term physical function and social participation. Medical treatments vary but typically consist of more than six months of intensive chemotherapy with surgical tumor resection performed approximately three months into treatment. Radiation therapy may be used in tumors that are responsive to radiation (such as Ewing sarcoma) or that may not be resectable. The most common antineoplastic chemotherapy agents for bone sarcoma are methotrexate, doxorubicin, cisplatin, vincristine, and cyclophosphamide [113,114]. Additionally, etoposide and ifosfamide are used in combination with the other chemotherapy agents or as a second-line treatment for recurrent or refractory disease [113,114]. These components of the medical intervention affect both cancerous and normal cells and cause short- and long-term side effects. Short-term side effects include nausea, vomiting, hair loss, and myelosuppression. Long-term effects include changes in the musculoskeletal (muscle weakness, decreased bone mineral density, asymmetry), neuromuscular (peripheral neuropathy), cardiopulmonary (increased risk for cardiovascular disease), and endocrine systems (short stature, diabetes mellitus, obesity) [115,116,117,118]. Additionally, specific cytotoxic chemotherapy agents, such as doxorubicin, have been associated with muscle atrophy, weakness, and impaired excitation–contraction coupling via downstream effects of elevated reactive oxygen species, alpha-tumor necrosis factor and mitochondrial dysfunction [8,9].

In addition to chemotherapy, surgical management consists of limb amputation or resection of the tumor with extensive reconstruction, known as limb-sparing surgery (LSS). LSS is the most common surgical procedure for local control of MSS and requires bone and/or soft tissue removal, reconstruction of resected bone segments, and muscular or tendon re-routing [119]. This is necessary to ensure the cancerous tissue is adequately resected and functional potential is maintained in the limb. The resected bone segments are reconstructed by endoprosthesis, hardware, autograft, and/or allograft placement. In addition to complex surgical techniques, considerations of growth are necessary in the management of children with immature musculoskeletal systems to prevent further secondary complications such as leg length discrepancies and biomechanical imbalances in muscle length and tension due to muscle re-routing [119]. Therefore, survivors of childhood MSS may present with impaired muscle strength and muscle properties, including smaller muscle size and lower neuromuscular activation of surgical and non-surgical limbs.

In survivors of childhood MSS of the lower extremities, muscle strength deficits are common. Tsauo et al. (2006) [10] reported ratios of muscle strength between the surgical and non-surgical limbs ranging from 37.4 to 47.5% for knee extension and 54.5 to 71.7% for knee flexion in 20 patients aged 13 to 40 years old who were treated for osteosarcoma proximal or distal to the knee joint, underwent limb-sparing total knee reconstruction, and completed chemotherapy. Muscle strength deficits are not only identified in the limb of the primary tumor but also have been identified in the non-surgical limb [11,12]. A case series by Beebe et al. (2009) [11] examined four skeletally immature children aged 9 to 11 years treated for MSS around the knee, underwent limb-sparing total knee reconstruction greater than one year prior, and received endoprosthesis lengthening. These same children demonstrated decreased non-operative knee and hip muscle force production compared to normative values for knee flexion (74%), knee extension (63%), hip flexion (35%), and hip extension (13%) [11]. Corr et al. (2017) [12] also found ankle dorsiflexion, hip flexion, and knee extension strength deficits in the non-operative limb in 13 children treated for MSS with an average age of 13.5 years. These strength deficits declined from baseline to post-surgery and did not recover by 20 to 22 weeks post-surgery [12]. Compared to a control group, Fernandez-Pineda et al. (2017) found significant impairment in MSS in maximal isokinetic knee extension and ankle plantarflexion and dorsiflexion joint torque when assessing the averaged peak value of the surgical and non-surgical limb [120]. In addition to skeletal muscle strength impairments, survivors of childhood MSS experience gross motor performance limitations such as spatiotemporal gait dysfunction, participation restrictions, and adverse general health affecting the quality of life [10,11,40,120,121,122,123,124,125,126,127,128,129,130,131,132,133,134,135,136,137]. One-third of survivors of MSS report high levels of physical limitations, a quarter reported moderate to severe limitations in daily tasks (25%), and survivors of MSS are more likely to report a decreased ability to perform personal care and routine activities [129,132,134,135,137,138,139].

There is evidence that survivors of childhood MSS have impairments in strength that contribute to gross motor performance deficits, such as gait dysfunction, after surgical management. Carty et al. (2009) [128] examined gait characteristics in 20 individuals with osteosarcoma after limb-sparing surgery greater than 1-year after their procedure and found that the amount of soft tissue removal during surgery, knee extension strength, and knee flexion range of motion predicted changes in gait patterns that reduced surgical limb knee and hip biomechanical demands. However, further exploration of contributing factors to muscle strength such as muscle size and neuromuscular activation and their relationships to gross motor performance is needed in survivors of childhood MSS.

## 6. Summary

The assessment of lower extremity muscle strength and its underlying mechanisms of muscle size and neuromuscular activation can provide valuable information about skeletal muscle performance in children, adolescents, and young adults with and without chronic health conditions. Clinicians and researchers should establish and implement measurements of skeletal muscle strength, size, and gross motor performance to assess changes in skeletal muscle related to health conditions and changes due to exercise interventions.

Comprehensive assessment of muscle performance, muscle size, neuromuscular activation, and gross motor performance is needed in clinical pediatric hematologic and oncologic populations. Survivors of childhood SCD and MSS undergo prolonged periods of medical and symptom management that increases inflammation and increases sedentary behaviors [8,140,141]. Due to the pathophysiological effects of SCD and MSS on skeletal muscle, these populations would significantly benefit from further exploration of skeletal muscle (Figure 4). The current body of literature exploring skeletal muscle in survivors of childhood SCD and MSS is just touching the surface. In order to transform rehabilitation referral, screening, and interventions, a comprehensive study of skeletal muscle and the underlying mechanisms that affect muscle performance and gross motor performance is needed. The enhanced knowledge and clinical applications of the measurements of muscle performance, muscle size, neuromuscular activation and identification of their relationships to gross motor performance are essential to advance rehabilitative care for children with hematological and oncological health conditions.

## Figures and Tables

**Figure 1 children-10-00114-f001:**
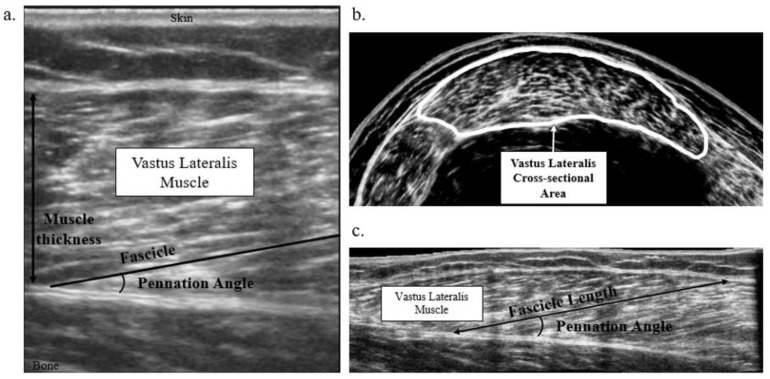
Ultrasonography images of the vastus lateralis muscle. (**a**) static long-axis image identifying muscle thickness, fascicle, and pennation angle; (**b**) extended-field-of-view short axis image identifying muscle anatomical cross-sectional area; (**c**) extended-field-of-view long axis image identifying fascicle length and pennation angle.

**Figure 2 children-10-00114-f002:**
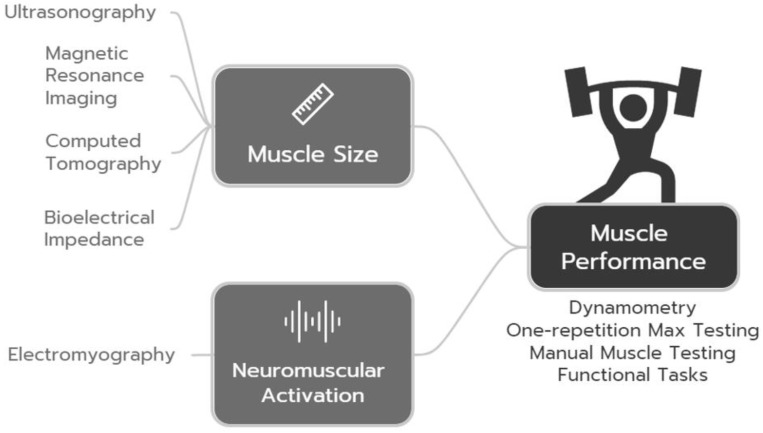
Assessment techniques to measure muscle performance.

**Figure 3 children-10-00114-f003:**
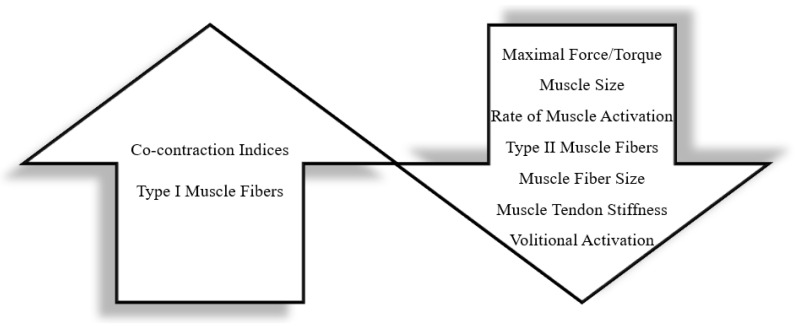
Expected Child–Adult Differences in Skeletal Muscle. The upward arrow includes elements of skeletal muscle that are expected to be increased in children compared to adults. The downward arrow includes elements of skeletal muscle that are expected to be decreased in children compared to adults.

**Figure 4 children-10-00114-f004:**
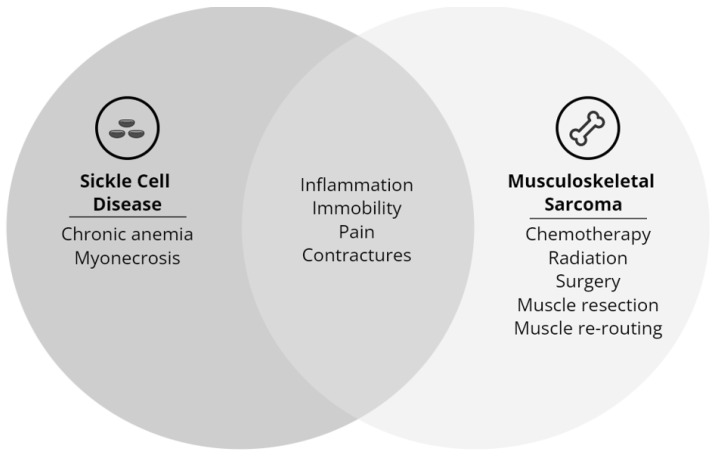
Underlying causes of skeletal muscle impairment in children with sickle cell disease (left), musculoskeletal sarcoma (right), and common causes across both diseases (center).

## Data Availability

Not applicable.

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
