# Peer review of "Skeletal Muscle Measurements in Pediatric Hematology and Oncology: Essential Components to a Comprehensive Assessment"

_children, 2023, doi:10.3390/children10010114_

Round 1

Reviewer 1 Report

 In this manuscript, the authors showed that skeletal muscle strength, size and neuromuscular activation indicate performance of muscle. In children with sickle cell disease or musculoskeletal sarcoma, the measurements of muscle performance are important for appropriate rehabilitation care. This manuscript is well organized. However, the following minor issue need authors’ attention.

 The authors described the measurement of muscle size by non-invasive imaging methods as an indicator of muscle performance. As they mentioned, muscle strength was examined in many previous reports, not its size. If possible, they had better show the concrete numeric about correlation between the size and strength, or describe age-specific average muscle size.

Author Response

Comment 1: In this manuscript, the authors showed that skeletal muscle strength, size and neuromuscular activation indicate performance of muscle. In children with sickle cell disease or musculoskeletal sarcoma, the measurements of muscle performance are important for appropriate rehabilitation care. This manuscript is well organized. However, the following minor issue need authors’ attention.

The authors described the measurement of muscle size by non-invasive imaging methods as an indicator of muscle performance. As they mentioned, muscle strength was examined in many previous reports, not its size. If possible, they had better show the concrete numeric about correlation between the size and strength, or describe age-specific average muscle size.

Response: We have included specific correlation/regression values to describe the associations between muscle size and strength in healthy children and children with cerebral palsy.  Please see lines 194-203.

Reviewer 2 Report

This review comprehensively evaluates muscle strength and its changes through pathological conditions such as sickle cell disease (SCD) and musculoskeletal neoplasms, a field with scarce data related to mechanisms that cause this observed phenomenon in patients. The main contribution and strength of this review are to coordinate data about muscle fundamentals such as physio-anatomical description, properties, functions, and techniques to measure muscle performance with association to diseases and their consequences and suggest a close follow-up of outcomes associated with muscle strength and quality of life-related to simple tasks such as walk, especially related to their development from a child through adult life.

In the review of Piel et al. (2017) they described that nearly every single organ can be affected by sickle cell disease (SCD). However, there is little information about how the muscular system is affected, with avascular necrosis and leg ulceration being the most frequent. Also on the CDC site, little information is found about this topic. Tanner et al (2020) argued about muscle strength in patients with cancer and their rehabilitation, and they debated the musculoskeletal impairments across consequences of cancer rehabilitation or treatment like chemotherapy or radiotherapy.

Scarce information about skeletal muscle strength, size, and neuromuscular activation are discussed in papers about patients with SCD, how their muscles develop through the years, and whether there is any loss of muscle capacity. In this context, some important works such as Gouraud et al. (2021), Marinho et al. (2016), Sachdev et al. (2016), and Waltz et al. (2013) who studied different populations with SCD and observed muscle alterations with no clear explanation concerning which phenomena are relevant were not cited in this review. All these works described correlations between some of the tests employed with SCD patients, with 6 minutes walking test being the consensus test applied by all those groups.

Gouraud et al., (2021) studied African patients with sickle cell disease (19 AA, 24SS e 18 SC) and they performed three isometric maximal voluntary contractions (iMVC) to determine Tmax of the quadriceps of the dominant leg, 6-Minute Walk Test (6-MWT), and Electromyographic activity (EMG) of the Vastus Lateralis (VL) using surface electrodes. Tmax and hematocrit were not independent predictors of the 6-MWT distance in SS and SC separately in their study, although both parameters should still be considered in the rehabilitation of SCD patients as they are closely related to the choice of step frequency. Waltz et al. (2013) and Marinho et al. (2016) found a positive relationship between the 6-MWT performance and the level of anemia in SS patients. Marinho et al. (2016) observed a significant correlation (P<0.001) between 6MWD and hemoglobin (Hb) level, tricuspid regurgitation velocity, forced vital capacity, acute chest syndrome, and diffusing capacity for carbon monoxide. Sachdev et al. (2011) observed a correlation between BMI and 6 MWDT and SCD patients and their findings do not fully explain PH, LVDD, and limited 6-minute walk distance in sickle cell anemia.

Besides most of those studies being done in the adult population, I still recommend some comments about those results being included in this review that have an emphasis on the child population to a wider overview in this field, approaching more comprehensive and comparative data in a subject that has limited information.

I still have minor suggestions about the format, concerning figure 3, which should be indicated which arrow is related to child and which is about adult muscle.

Author Response

Comment:  Gouraud et al., (2021) studied African patients with sickle cell disease (19 AA, 24SS e 18 SC) and they performed three isometric maximal voluntary contractions (iMVC) to determine Tmax of the quadriceps of the dominant leg, 6-Minute Walk Test (6-MWT), and Electromyographic activity (EMG) of the Vastus Lateralis (VL) using surface electrodes. Tmax and hematocrit were not independent predictors of the 6-MWT distance in SS and SC separately in their study, although both parameters should still be considered in the rehabilitation of SCD patients as they are closely related to the choice of step frequency.

Response: This article by Gouraund et al. (2021) identified positive relationships between Tmax and 6-MWT distance in participants with SS-genotype SCD, yet no relationships with EMG measurements. We added these important new findings in adults with SCD to lines 413-416.

Comment: Waltz et al. (2013) and Marinho et al. (2016) found a positive relationship between the 6-MWT performance and the level of anemia in SS patients. Marinho et al. (2016) observed a significant correlation (P<0.001) between 6MWD and hemoglobin (Hb) level, tricuspid regurgitation velocity, forced vital capacity, acute chest syndrome, and diffusing capacity for carbon monoxide. Sachdev et al. (2011) observed a correlation between BMI and 6 MWDT and SCD patients and their findings do not fully explain PH, LVDD, and limited 6-minute walk distance in sickle cell anemia.

Besides most of those studies being done in the adult population, I still recommend some comments about those results being included in this review that have an emphasis on the child population to a wider overview in this field, approaching more comprehensive and comparative data in a subject that has limited information.

Response: We agree that these manuscripts provide important information about underlying mechanisms that may contribute to gross motor and exercise performance in children and adults with SCD, however, the direct links to skeletal muscle are not apparent. A comprehensive review of physical impairments and function in children and adolescents with SCD have been reported in Marchese et al. (2022). We do agree that in the future studies exploring these direct links are needed.

Comment: I still have minor suggestions about the format, concerning figure 3, which should be indicated which arrow is related to child and which is about adult muscle.

Response: Lines 292-295:Figure 3. Expected Child-Adult Differences in Skeletal Muscle. The upward arrow includes elements of skeletal muscle that are expected to be increased in children compared to adults. The downward arrow includes elements of skeletal muscle that are expected to be decreased in children compared to adults.” -please let us know if this caption requires further clarification.